# HyperET: Efficient Training in Hyperbolic Space for Multi-modal Large Language Models

**Zelin Peng[1], Zhengqin Xu[2], Qingyang Liu[1], Xiaokang Yang[1], Wei Shen[1] (✉)**

[1] MoE Key Lab of Artificial Intelligence, AI Institute, School of Computer Science, SJTU
[2] State Key Laboratory of Infrared Physics, Shanghai Institute of Technical Physics, CAS
{zelin.peng, fate311, narumimaria, xkyang, wei.shen}@sjtu.edu.cn

## Abstract

Multi-modal large language models (MLLMs) have emerged as a transformative approach for aligning visual and textual understanding. They typically require extremely high computational resources (e.g., thousands of GPUs) for training to achieve cross-modal alignment at multi-granularity levels. We argue that a key source of this inefficiency lies in the vision encoders they widely equip with, e.g., CLIP and SAM, which lack the alignment with language at multi-granularity levels. To address this issue, in this paper, we leverage hyperbolic space, which inherently models hierarchical levels and thus provides a principled framework for bridging the granularity gap between visual and textual modalities at an arbitrary granularity level. Concretely, we propose an efficient training paradigm for MLLMs, dubbed as HyperET, which can optimize visual representations to align with their textual counterparts at an arbitrary granularity level through dynamic hyperbolic radius adjustment in hyperbolic space. HyperET employs learnable matrices with Möbius multiplication operations, implemented via three effective configurations: diagonal scaling matrices, block-diagonal matrices, and banded matrices, providing a flexible yet efficient parametrization strategy. Comprehensive experiments across multiple MLLM benchmarks demonstrate that HyperET consistently improves both existing pre-training and fine-tuning MLLMs clearly with less than 1% additional parameters. Code is available at https://github.com/godlin-sjtu/HyperET.

## 1   Introduction

Thanks to advancements in pre-trained foundation models in both computer vision and natural language processing [60, 61, 58, 1, 29, 50, 14, 18, 50, 15], researchers have been inspired to explore the alignment of visual and language models, leading to the development of multi-modal large language models (MLLMs). This alignment is often achieved through adapters, e.g., Q-former [16]. As a result, MLLMs [2, 10, 16, 40, 49, 4, 9, 65] rapidly develop in recent years, demonstrating strong performance in tasks that require both visual and textual understanding, e.g., image captioning and visual question answering (VQA).

Although modern multimodal large language models (MLLMs), e.g., Qwen-VL series [5, 64, 4] and Intern-VL series [13, 12, 11], achieve cross-modal alignment across a wide range of MLLM tasks that inherently involve multi-granularity levels, their success heavily depends on extensive data scaling and massive computational resources. For example, InternVL [13] requires training on hundreds of millions of image-text pairs using up to 640 GPUs. Such a resource-intensive training scheme raises serious concerns about efficiency, reproducibility, and long-term sustainability in the MLLM

---

✉ Corresponding Author: wei.shen@sjtu.edu.cn

39th Conference on Neural Information Processing Systems (NeurIPS 2025).

community, especially for researchers and institutions constrained by limited computational resources. To address these concerns, it is essential to identify the underlying cause of this inefficiency. We argue that a key factor lies in the vision encoders that are commonly used, e.g., CLIP [54], SAM [29], and DINOv2 [46]. These encoders are typically aligned with language at a single granularity level, e.g., either pixel-level or object-level, and are thus insufficient to deal with tasks required for alignments at different granularity levels. This mismatch in granularity during training significantly impedes the optimization process, leading to inefficient cross-modal alignment and increased reliance on large-scale computational resources.

To solve this issue, we propose to directly quantify the granularity levels by leveraging hyperbolic space [7]. Empirical observations in prior works demonstrate that visual representations at different hierarchical levels (e.g., image-level and object-level) naturally stratified in hyperbolic space [20, 52, 47]. This property enables the use of hyperbolic radius [57]—defined as the distance from a point to the origin in hyperbolic space—to quantify granularity levels [47]. Specifically, points in hyperbolic space closer to the origin (smaller radius) encode low-level visual features (e.g., pixel-level information), while points near the boundary (larger radius) represent high-level visual semantics (e.g., image-level concepts). This hierarchical level suggests that adjusting the hyperbolic radius of visual representations can effectively align them with language models at arbitrary granularity levels.

Building on this insight, we propose an efficient training paradigm (HyperET) for MLLMs that is capable of optimizing visual representations to align with their textual counterparts at arbitrary granularity levels. This is achieved through learnable matrices equipped with Möbius multiplication operations [62], which enable direct and continuous adjustment of the hyperbolic radius of visual representations, thereby facilitating cross-modal alignment. In practice, we introduce three parameter-efficient forms of the learnable matrices for adjusting the hyperbolic radius: (1) diagonal scaling matrices, (2) block-diagonal scaling matrices, and (3) banded scaling matrices. These designs significantly reduce the number of trainable parameters while retaining the capacity to address granularity mismatch. To further enhance parameter flexibility, the matrices can be extended to a dense version with fully populated learnable elements. This expansion increases the expressive capacity of our proposed training paradigm, facilitating more effective alignment across diverse cross-modal scenarios.

To evaluate the generalization capability and effectiveness of HyperET, we conduct extensive experiments across multiple pre-trained MLLM benchmarks and various downstream MLLM tasks, e.g., ScienceQA [42]. Results demonstrate that the proposed HyperET can be easily plugged-and-play and consistently improve various MLLMs, including LLaVA-1.5 [39] and LLaVA-Next [32] for pre-training, as well as MemVP [26] and LaVIN [43] for fine-tuning on downstream tasks. Notably, HyperET introduces less than 1% of the total trainable parameters, ensuring high parameter efficiency.

## 2 Related Work

### 2.1 Multi-modal Large Language Models

Multi-modal large language models (MLLMs) [31, 10, 33, 38, 2, 55, 36, 70] make significant breakthroughs in recent advancements, aiming to equip large language models [73, 1, 60, 61, 3] with the capability to process and interpret visual information. Most MLLMs achieve this goal by integrating a CLIP's vision encoder [69, 54] into pre-trained large language models through adapters, e.g., MLPs [40, 39], Q-Former [16, 35], and attention mechanisms [2]. Despite its effectiveness, this straightforward connection between the visual and language modalities still fails to align pre-trained models from different modalities, thereby resulting in inferior performance, e.g., hallucinations, across various downstream tasks.

### 2.2 Towards Alignment in MLLM

This failure mainly stems from the fact that the CLIP's vision encoder [50] is designed for standard classification tasks and is not equipped to handle the more fine-grained visual understanding tasks required by the language modality [59, 51, 27]. To better align the vision and language modalities in MLLMs, recent works explore parameter-efficient fine-tuning methods [43, 56, 71, 26, 44]. For example, LaVIN [43] introduces adapters in both the vision encoder and LLaMA [60] to achieve better alignment of the modalities. Another line of research [66, 59, 27] seeks to overcome this

bottleneck by incorporating additional vision models, such as DINOv2 [46], to construct a more powerful and capable vision branch. Despite the advancements in the field, few studies focus on understanding the changes in the representation of vision encoders that enable MLLMs to tackle more complex vision tasks. In this work, we aim to bridge this gap by leveraging hyperbolic space to directly model the granularity levels and adapt the granularity of visual representations to an appropriate level.

## 2.3 Learning in Hyperbolic Space

Unlike Euclidean space, hyperbolic space can be viewed as the continuous analog of a tree [6], making it inherently suitable for capturing hierarchical levels among various data types. Since visual and textual concepts are inherently hierarchical, recent works [17, 30, 52, 47, 48] show that hyperbolic space serves as a promising manifold for preserving granularity levels in vision-language model representations, leading to strong performance across downstream tasks. Most existing methods directly reshape the original hierarchical structure to align different modalities. In contrast, our method preserves the original hierarchical structure and specifically leverages the intrinsic property of hyperbolic radius to enable visual representations to adapt their hierarchical level, aligning them with the language modality. This approach provides a complementary perspective to existing efforts in the MLLM community.

## 3 Preliminary Concepts

**Hyperbolic Geometry.** In contrast to Euclidean or spherical geometries, hyperbolic geometry is characterized by a constant negative curvature, which fundamentally distinguishes its geometric properties and computational behaviors. Following prior studies [8, 57], we utilize the classical Poincaré ball model—one of five principal analytic models for constructing hyperbolic space [7]—due to its demonstrated efficacy in representing hierarchical levels [23, 19, 45]. The Poincaré ball model $(\mathbb{D}_c^n, g^{\mathbb{D}_c})$, characterized by a radius of $1/\sqrt{c}$ and constant negative curvature $-c$ ($c > 0$) is formally defined as follows:

$$\begin{cases} \mathbb{D}_c^n := \{\mathbf{X} \in \mathbb{R}^n : c\|\mathbf{X}\| < 1\} \\ g^{\mathbb{D}_c} := \lambda_{c,\mathbf{X}}^2 g^E \end{cases}, \tag{1}$$

where $\lambda_{c,\mathbf{X}} = \frac{2}{1-c\|\mathbf{X}\|^2}$ and $g^E = \mathbf{I}_n$ denotes the Euclidean metric tensor, serving as the foundation for the hyperbolic space construction.

**Hyperbolicity.** Building upon the theoretical framework of gyrovector spaces [62, 63], we incorporate Möbius operations into hyperbolic space, specifically the Möbius addition operation "$\oplus_c$" and Möbius multiplication operation "$\otimes_c$". In hyperbolic geometry, the tangent space $\mathcal{T}_{\mathbf{X}}^c \mathbb{D}_c^n$ at any point $\mathbf{X} \in \mathbb{D}_c^n$ serves as a first-order approximation of $\mathbb{D}_c^n$, representing an $n$-dimensional Euclidean space that locally approximates the hyperbolic structure. The tangent space $\mathcal{T}_{\mathbf{X}}^c \mathbb{D}_c^n$ and $\mathbb{D}_c^n$ are mapped to each other by exponential ($\mathcal{T}_{\mathbf{X}}^c \mathbb{D}_c^n \mapsto \mathbb{D}_c^n : \exp_{\mathbf{X}}^{\mathbb{D},c}(\cdot)$) and logarithmic ($\mathbb{D}_c^n \mapsto \mathcal{T}_{\mathbf{X}}^c \mathbb{D}_c^n : \log_{\mathbf{X}}^{\mathbb{D},c}(\cdot)$) maps, respectively. Detailed mathematical definitions and derivations are provided in the supplementary material.

## 4 Methodology

This section first provides the necessary background on the conventional training paradigm from the perspective of parameter space tuning (Sec. 4.1) to contextualize our approach, followed by the detailed presentation of HyperET in Sec. 4.2. Then, Sec. 4.3 provides a theoretical analysis on adjusting the hyperbolic radius of visual representations.

### 4.1 Parameter Space Tuning

Parameter space tuning in multi-modal large language models (MLLMs), whether through full fine-tuning or parameter-efficient fine-tuning, aims to adapt pre-trained visual and language models to target multi-modal scenarios. However, these tuning methods, which rely solely on gradient updates, operate as constraint-free adjustments in Euclidean space. They often implicitly assume that visual representations can sufficiently adapt to the required granularity level (e.g., transitioning

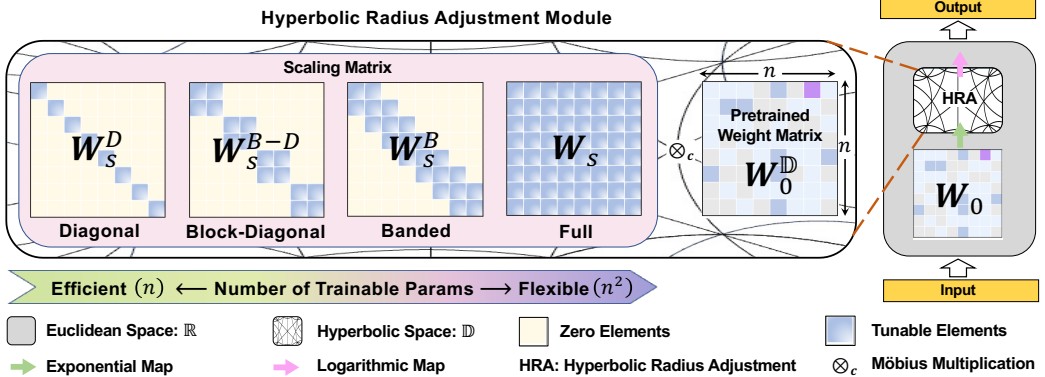

Figure 1: **The schematic representation of HyperET.** In HyperET, we efficiently train MLLMs in hyperbolic space by adjusting the hyperbolic radius using a tunable scaling matrix $\mathbf{W}_s$. Here, $\mathbf{W}_s$ can be configured into three parameter-efficient variants, i.e., Diagonal, Block-Diagonal and Banded.

from image-level to pixel-level), which may lead to inefficiency in alignment at a certain granularity level. In contrast, the core technique of our proposed HyperET involves hyperbolic radius adjustment, which explicitly adjusts the granularity level of visual representations in MLLMs. HyperET provides a simple yet effective solution to granularity mismatch challenges.

## 4.2 General Hyperbolic Radius Adjustment

As previously discussed, hyperbolic radius adjustment provides a direct mechanism for optimizing the granularity level of visual representations, effectively bridging the granularity gap between visual and language modalities. In practice, we introduce a radius adjustment constraint into the weight update process, generally defined as follows:

$$\text{Rad}_{\mathbf{W}^{\mathbb{D}}}/\text{Rad}_{\mathbf{W}_0^{\mathbb{D}}} = s \quad \Leftrightarrow \quad \text{Rad}_{\mathbf{W}^{\mathbb{D}}} = s \cdot \text{Rad}_{\mathbf{W}_0^{\mathbb{D}}}, \tag{2}$$

where $\text{Rad}_{\mathbf{W}^{\mathbb{D}}}$ and $\text{Rad}_{\mathbf{W}_0^{\mathbb{D}}}$ represent the hyperbolic radii of $\mathbf{W}^{\mathbb{D}}$ and $\mathbf{W}_0^{\mathbb{D}}$, respectively, quantifying their granularity levels. The hyperbolic weight matrices $\mathbf{W}^{\mathbb{D}}$ and $\mathbf{W}_0^{\mathbb{D}}$ are derived by projecting their Euclidean counterparts $\mathbf{W}$ and $\mathbf{W}_0$ into hyperbolic space through exponential mapping operations defined in Sec. 3. The scaling coefficient $s$ is task-adaptive, dynamically adjusting to optimize performance. However, this modification requires a two-step procedure: (1) computing the hyperbolic radius of $\mathbf{W}^{\mathbb{D}}$ and (2) subsequently adjusting it. To streamline this process, we propose a more efficient approach that directly optimizes $\mathbf{W}^{\mathbb{D}}$ without intermediate radius computations.

**Hyperbolic Radius.** Without loss of generality, for any point $\mathbf{X} \in \mathbb{D}_c^n$ in hyperbolic space, its hyperbolic radius is formally defined as follows:

$$\text{Rad}_{\mathbf{X}} := d_c^{\mathbb{D}}(\mathbf{X}, \mathbf{0}) = (\frac{2}{\sqrt{c}})\tanh^{-1}(\sqrt{c}\|\mathbf{X}\|), \tag{3}$$

where $\mathbf{0}$ denotes the origin point in hyperbolic space. In the subsequent analysis, we demonstrate that Mobiüs multiplication operations $\otimes_c$ defined in Sec. 3 can enable precise control over the hyperbolic radius. This critical property is formally stated in the following theorems:

**Theorem 1** (Hyperbolic Radius Scaling) For a point $\mathbf{X} \in \mathbb{D}_c^n$ in hyperbolic space, the hyperbolic radius adjustment function is expanded as follows:

$$\begin{aligned}
s \cdot \text{Rad}_{\mathbf{X}} &= \frac{2}{\sqrt{c}}(s\frac{\sqrt{c}}{2}\text{Rad}_{\mathbf{X}}) \\
&= \frac{2}{\sqrt{c}}\tanh^{-1}(\sqrt{c}\|s \otimes_c \mathbf{X}\|) \\
&= \text{Rad}_{s \otimes_c \mathbf{X}}.
\end{aligned} \tag{4}$$

where $\otimes_c$ here is instantiated as a Mobiüs scalar multiplication operation. Therefore, hyperbolic radius adjustment can be precisely controlled through $\otimes_c$ between $s$ and $\mathbf{X}$, with the scaling coefficient $s$ serving as a primary learnable parameter. ∎

According to Theorem 1, $\otimes_c$ can provide an equivalent mechanism during parameter space tuning. Consequently, hyperbolic radius adjustment in Eq. (2) can be easily achieved as follows:

$$\mathbf{W}^{\mathbb{D}} = s \otimes_c \mathbf{W}_0^{\mathbb{D}}. \tag{5}$$

Then, building upon the exponential mapping $\exp_{\mathbf{0}}^{\mathbb{D},c}(\cdot)$ and the logarithmic mapping $\log_{\mathbf{0}}^{\mathbb{D},c}(\cdot)$ defined in Sec. 3, we achieve general hyperbolic radius adjustment via the following reformulation of Eq. (5):

$$\mathbf{W} = \log_{\mathbf{0}}^{\mathbb{D},c}(s \otimes_c \exp_{\mathbf{0}}^{\mathbb{D},c}(\mathbf{W}_0)), \tag{6}$$

where $s$ is a learnable parameter for adjustment. Given that a constrained parameter set—particularly when limited to a single learnable parameter—may ineffectively adjust hyperbolic radius during restricted training iterations, we propose a more flexible parameterization strategy through matrix-based formulations, termed flexible adjustment, which further enables precise control over hyperbolic radius optimization.

**Flexible Adjustment for Hyperbolic Radius.** To achieve this, we adopt a scaling matrix $\mathbf{W}_s$ to replace the scaling coefficient $s$, which is satisfied: $s = \|\mathbf{W}_s \mathbf{W}_0^{\mathbb{D}}\|/\|\mathbf{W}_0^{\mathbb{D}}\|$. Consequently, the right-hand side of Eq. (2) can be reformulated as follows:

$$\begin{aligned}
\mathrm{Rad}_{\mathbf{W}^{\mathbb{D}}} &= s \cdot \mathrm{Rad}_{\mathbf{W}_0^{\mathbb{D}}} \\
&= \frac{\|\mathbf{W}_s \mathbf{W}_0^{\mathbb{D}}\|}{\|\mathbf{W}_0^{\mathbb{D}}\|} \cdot \mathrm{Rad}_{\mathbf{W}_0^{\mathbb{D}}}.
\end{aligned} \tag{7}$$

In this framework, the learnable parameters transition from scalar values to matrix-based formulations, significantly enhancing flexibility within constrained training iterations. The following theorem demonstrates that hyperbolic radius can be dynamically scaled through Mobiüs matrix multiplication operations, a generalized instantiation of $\otimes_c$.

**Theorem 2** (Hyperbolic Radius Flexibility Scaling) For a point $\mathbf{X} \in \mathbb{D}_c^n$ in hyperbolic space and a scaling matrix $\mathbf{X}_s \in \mathbb{R}^{n \times n}$, the flexible radius adjustment function is formally defined through Eq. (7) and Mobiüs matrix multiplication operations as follows:

$$\begin{aligned}
\frac{\|\mathbf{X}_s \mathbf{X}\|}{\|\mathbf{X}\|} \cdot \mathrm{Rad}_{\mathbf{X}} &= \frac{2}{\sqrt{c}}\left(\frac{\|\mathbf{X}_s \mathbf{X}\|}{\|\mathbf{X}\|}\frac{\sqrt{c}}{2}\mathrm{Rad}_{\mathbf{X}}\right) \\
&= \frac{2}{\sqrt{c}}\tanh^{-1}(\sqrt{c}\|\mathbf{X}_s \otimes_c \mathbf{X}\|) \\
&= \mathrm{Rad}_{\mathbf{X}_s \otimes_c \mathbf{X}}.
\end{aligned} \tag{8}$$

Analogically, the scaling matrix $\mathbf{W}_s$ is able to direct adjust the hyperbolic radius $\mathrm{Rad}_{\mathbf{X}}$ through Mobiüs matrix multiplication operations, providing precise control over representation granularity. ∎

Building upon Theorem 2, we introduce a matrix-based formulation by replacing the scaling coefficient $s$ with $\mathbf{W}_s$, resulting in the following reformulation of Eq. (6):

$$\mathbf{W} = \log_{\mathbf{0}}^{\mathbb{D},c}(\mathbf{W}_s \otimes_c \exp_{\mathbf{0}}^{\mathbb{D},c}(\mathbf{W}_0)). \tag{9}$$

With up to $O(n^2)$ learnable elements, $\mathbf{W}_s$ in Eq. (9) offers significantly enhanced flexibility for hyperbolic radius adjustment. Notably, Eqs. (5) and (9) become equivalent when $\mathbf{W}_s$ is constrained to a diagonal matrix with uniform scaling factors, i.e., $\mathbf{W}_s = \mathrm{diag}(\omega_1, \omega_2, \cdots, \omega_n) \in \mathbb{R}^{n \times n}$ and $\omega_i = \omega_j, i \neq j$. Moreover, in alignment with the current paradigm of parameter-efficient fine-tuning, we also introduce a parameter-efficient variant of $\mathbf{W}_s$, designed to maintain flexibility while reducing computational overhead.

**Efficient Adjustment for Hyperbolic Radius.** To realize this strategy, we define a *diagonal scaling matrix* $\mathbf{W}_s^D = \mathrm{diag}(\omega_1, \omega_2, \ldots, \omega_n) \in \mathbb{R}^{n \times n}$, where each $\omega_i$ is a learnable scalar and $\omega_i \neq \omega_j$ when $i \neq j$, significantly reducing the number of learnable parameters. In this form, only the diagonal elements are learnable parameters, making the diagonal scaling matrix $\mathbf{W}_s^D$ the most parameter-efficient configuration for hyperbolic radius adjustment. Building upon this foundation, we extend $\mathbf{W}_s^D$ into two more flexible variants by incrementally introducing learnable off-diagonal elements, balancing enhanced adjustment capability with parameter efficiency: (1) *Block-diagonal scaling matrix* $\mathbf{W}_s^{B\text{-}D} = \mathrm{diag}(\mathbf{R}_1, \mathbf{R}_2, \ldots, \mathbf{R}_r)$, where $\mathbf{R}_i \in \mathbb{R}^{\frac{n}{r} \times \frac{n}{r}}$. Here, $r$ is the block size (we assume

$n$ is divisible by $r$). This formulation allows intra-block interactions while maintaining sparsity across blocks; (2) *Banded scaling matrix* $\mathbf{W}_s^B = \begin{bmatrix} \omega_{11} & \cdots & \omega_{1n} \\ \vdots & \ddots & \vdots \\ \omega_{n1} & \cdots & \omega_{nn} \end{bmatrix} \in \mathbb{R}^{n \times n}$, where $\omega_{ij} = 0$ for all $i, j$ such that $|i - j| > d$. Here, $d$ denotes the bandwidth, indicating that nonzero elements are allowed within $d$ entries above and below the main diagonal. These two variants provide mechanisms to capture localized interactions while preserving parameter efficiency. Notably, both $\mathbf{W}_s^{B\text{-}D}$ and $\mathbf{W}_s^B$ degenerate to $\mathbf{W}_s^D$ under specific configurations: when $r = n$ for the block-diagonal matrix and $d = 0$ for the banded matrix. This highlights the inherent hierarchical flexibility enabled by our parametrization strategy and reflects a insightful way of designing fine-tuning methods.

To summarize these fine-tuning strategies, Fig. 1 provides a unified illustration of different variants of the scaling matrix $\mathbf{W}_s$. Our introduced hyperbolic radius adjustment, i.e., HRA, adjusts visual representations by applying learnable scaling matrices to the frozen pre-trained weights $\mathbf{W}_0$ in hyperbolic space, enabling precise adjustment over arbitrary granularity levels.

### 4.3 Theoretical Analysis

To demonstrate the impact of hyperbolic radius adjustment, we provide a theoretical analysis of the Möbius multiplication operation. The following deduction shows that the proposed Möbius multiplication operation can directly adjust the granularity level of visual representations.

**Deduction.** For $\mathbf{Y}_0 = \mathbf{W}_0 \mathbf{X}$, where $\mathbf{X} \in \mathbb{R}^{d \times k}$ is the input embedding, $\mathbf{W}_0$ is the pre-trained weight and $\mathbf{Y}_0$ is the pre-trained visual representation, the forward pass of HyperET is as follows:

$$\mathbf{Y}_0 = \mathbf{W}\mathbf{X} = \log_{\mathbf{0}}^{\mathbb{D},c}(\mathbf{W}_s \otimes_c \exp_{\mathbf{0}}^{\mathbb{D},c}(\mathbf{W}_0))\mathbf{X}. \tag{10}$$

Then, according to the definition of hyperbolic space, $\mathbf{Y}_0$ is firstly projected into the hyperbolic space $\mathbb{D}_c^n$ and our HyperET is applied to adjust the hyperbolic radius, resulting in the new visual representation $\mathbf{Y}$, expressed as:

$$\mathbf{Y}_0^{\mathbb{D}_c^n} = \exp_{\mathbf{0}}^{\mathbb{D},c}(\mathbf{Y}_0) = \mathbf{W}_0^{\mathbb{D}_c^n} \mathbf{X}^{\mathbb{D}_c^n} \tag{11}$$

$$\mathbf{Y}^{\mathbb{D}_c^n} = \exp_{\mathbf{0}}^{\mathbb{D},c}(\mathbf{Y}) = \mathbf{W}_s \otimes_c \mathbf{W}_0^{\mathbb{D}_c^n} \mathbf{X}^{\mathbb{D}_c^n}, \tag{12}$$

where $\mathbf{W}_0^{\mathbb{D}_c^n} = \exp_{\mathbf{0}}^{\mathbb{D},c}(\mathbf{W}_0)$ and $\mathbf{X}^{\mathbb{D}_c^n} = \exp_{\mathbf{0}}^{\mathbb{D},c}(\mathbf{X})$. According to the Theorem 2, the hyperbolic radius of $\mathbf{Y}^{\mathbb{D}_c^n}$ is obtained as:

$$\begin{aligned}
\text{Rad}_{\mathbf{Y}} &= \text{Rad}_{\mathbf{W}_s \otimes_c \mathbf{W}_0^{\mathbb{D}_c^n} \mathbf{X}^{\mathbb{D}_c^n}} \\
&= \frac{\|\mathbf{W}_s \mathbf{W}_0^{\mathbb{D}_c^n} \mathbf{X}^{\mathbb{D}_c^n}\|}{\|\mathbf{W}_0^{\mathbb{D}_c^n} \mathbf{X}^{\mathbb{D}_c^n}\|} \cdot \text{Rad}_{\mathbf{W}_0^{\mathbb{D}_c^n} \mathbf{X}^{\mathbb{D}_c^n}} \\
&= \frac{\|\mathbf{W}_s \mathbf{W}_0^{\mathbb{D}_c^n} \mathbf{X}^{\mathbb{D}_c^n}\|}{\|\mathbf{W}_0^{\mathbb{D}_c^n} \mathbf{X}^{\mathbb{D}_c^n}\|} \text{Rad}_{\mathbf{Y}_0} \\
&= s \cdot \text{Rad}_{\mathbf{Y}_0}, \tag{13}
\end{aligned}$$

where $s$ is a scaling coefficient. Therefore, our hyperbolic radius adjustment method can directly adjust the hyperbolic radius of visual representations, thus is able to bridge the granularity gap between visual and textual modalities at an arbitrary granularity level.

## 5 Experiments

Our experimental evaluation encompasses two MLLM scenarios, (1) MLLM's fine-tuning (Sec. 5.1), and (2) MLLM's pre-training (Sec. 5.2). A detailed ablation study is presented in Sec. 5.3 and 5.4.

### 5.1 MLLM's Fine-tuning

**Experimental Setting.** We evaluate our method on ScienceQA [42], a challenging large-scale VQA benchmark encompassing diverse scientific domains. Our comparative analysis includes MemVP and other LLaMA-based models with input-space visual prompting: LLaVA [40], and LaVIN [43]. We

Table 1: **Comparision with SoTA fine-tuning methods** on ScienceQA test set [42]. Question categories: NAT = natural science, SOC = social science, LAN = language science, TXT = w/ text context, IMG = w/ image context, NO = no context, G1-6 = grades 1-6, G7-12 = grades 7-12. "Ours": we here realize the extra learnable parameters as diagonal matrices, i.e., $\mathbf{W}_s^D$. Vision encoder: CLIP.

| Method | #Trainable Params | Language Model | Subject | | | Context Modality | | | Grade | | Average |
|---|---|---|---|---|---|---|---|---|---|---|---|
| | | | NAT | SOC | LAN | TXT | IMG | NO | G1-6 | G7-12 | |
| Human | - | - | 90.23 | 84.97 | 87.48 | 89.60 | 87.50 | 88.10 | 91.59 | 82.42 | 88.40 |
| *Fully Fine-Tuning* | | | | | | | | | | | |
| LLaVA | 13B | Vicuna-13B | 90.36 | **95.95** | 88.00 | 89.49 | 88.00 | 90.66 | 90.93 | **90.90** | 90.92 |
| *Parameter-efficient Fine-Tuning* | | | | | | | | | | | |
| LaVIN | 3.8M | LLaMA-7B | 89.25 | 94.94 | 85.24 | 88.51 | 87.46 | 88.08 | 90.16 | 88.07 | 89.41 |
| LaVIN+Ours | 3.85M (+0.05M) | LLaMA-7B | **89.35** | **96.06** | **86.54** | 88.29 | **88.01** | **89.33** | **91.36** | 87.65 | **90.03** (+0.62) |
| MemVP | 3.9M | LLaMA-7B | 94.45 | 95.05 | 88.64 | 93.99 | 92.36 | 90.94 | 93.10 | 93.01 | 93.07 |
| MemVP+Ours | 3.95M (+0.05M) | LLaMA-7B | **94.85** | 95.05 | **90.55** | **94.57** | **92.91** | **92.20** | **93.65** | **94.00** | **93.78** (+0.71) |
| LaVIN | 5.4M | LLaMA-13B | 90.32 | 94.38 | 87.73 | 89.44 | 87.65 | 90.31 | 91.19 | 89.26 | 90.50 |
| LaVIN+Ours | 5.45M (+0.05M) | LLaMA-13B | **90.57** | **95.63** | **89.89** | **89.61** | **88.75** | **92.02** | **91.95** | **90.58** | **91.46** (+0.96) |
| MemVP | 5.5M | LLaMA-13B | 95.07 | 95.15 | 90.00 | 94.43 | 92.86 | 92.47 | 93.61 | 94.07 | 93.78 |
| MemVP+Ours | 5.55M (+0.05M) | LLaMA-13B | **96.19** | **95.78** | **90.86** | **95.51** | **94.25** | **93.18** | **94.88** | **94.44** | **94.72** (+0.94) |

follow the experiment setting in [43]. All models utilize a CLIP pre-trained ViT-L/14 visual encoder. The weights of HyperET in this task are implemented using the three parameter-efficient scaling matrices, i.e., $\mathbf{W}_s^D$, $\mathbf{W}_s^{B-D}$ and $\mathbf{W}_s^B$, and are adapted in the attention layer, consistent with most parameter-efficient tuning methods, e.g., LoRA [24]. The curvature $c$ is 0.01. All experiments are conducted using a maximum of 8 NVIDIA H800 GPUs.

**Comparing to SOTA.** Our experimental evaluation compares the proposed approach with state-of-the-art parameter-efficient fine-tuning (PEFT) methods, including LaVIN [43] and MemVP [26]. As demonstrated in Table 4, which presents both baseline and HyperET enhanced results, our method establishes new state-of-the-art performance. HyperET achieves this breakthrough with minimal parameter overhead (fewer than 1%), delivering substantial improvements to both LaVIN and MemVP frameworks. Particularly noteworthy are the gains observed with LLaMA-13B as the backbone language model, where HyperET enhances average cross-domain accuracy by 0.96% for LaVIN and 0.94% for MemVP. Notably, when integrated with MemVP using LLaMA-7B, HyperET achieves performance on par with the more computationally intensive MemVP-LLaMA-13B configuration, i.e., 93.78. This result demonstrates that HyperET enhances visual representation and achieves comparable performance improvements while utilizing $100,000\times$ fewer parameters (0.05M vs 6B) than MemVP using LLaMA-13B, highlighting its remarkable parameter efficiency.

## 5.2 MLLM's Pre-training

**Experimental Setting.** Our pre-training evaluation framework builds upon LLaVA-1.5 [39], employing identical datasets to evaluate HyperET's effectiveness in MLLM pre-training. Our comparative analysis includes LLaVA-1.5 [39] and LLaVA-Next [39] and train and fine-tune our HyperET with the same experiment setting. The weights of HyperET in this task are implemented using the highest flexible scaling matrices, i.e., $\mathbf{W}_s$, and are adapted in the attention layer, consistent with most parameter-efficient training methods, e.g., LoRA [24].

**Comparing to SOTA.** Our experimental framework evaluates the proposed method against state-of-the-art pre-trained MLLMs across 12 standard visual language benchmarks. We implement HyperET using the most flexible matrix configuration, i.e., $\mathbf{W}_s$, maximizing the model's adaptability. While this approach resembles full fine-tuning in structure, the actual parameter count remains remarkably efficient at approximately 50M—less than 1% of the language model's 13B parameters—maintaining parameter efficiency. As shown in Table 2, our approach demonstrates significant performance improvements over LLaVA-1.5 [39], particularly in mitigating the limitations of CLIP-based encoders. Notably, on the POPE benchmark [37] for object hallucination detection, HyperET substantially reduces visual hallucinations, providing empirical evidence that hyperbolic radius optimization effectively enhances the cross-modal alignment at an arbitrary level.

Table 2: **Comparison with SoTA pre-trained methods** on 12 MLLM benchmarks, including VQAv2 [21], GQA [25], VW: VisWiZ [22], SQA: ScienceQA-IMG [42], TVQA: TextVQA [53], PE: POPE [37], ME: MME [67], MB: MMBench [41], MB$^{CN}$: MMBench-Chinese [41], SD: SEED-Bench [34], LVA$^W$: LLaVA-Bench (In-the-Wild) [40] and M-Vet [68]. Top-1 accuracy is reported (Best in **bold**, second best is underlined). Lan. Model: Language model. Benchmark names are abbreviated due to space limits. "Ours": we here realize the extra learnable parameters as full matrices, i.e., $\mathbf{W}_s$. Vision encoder: CLIP.

| Method | Lan. Model | VQAv2 | GQA | VW | SQA | TVQA | PE | ME | MB | MB$^{CN}$ | SD | LVA$^W$ | M-Vet |
|---|---|---|---|---|---|---|---|---|---|---|---|---|---|
| LLaVA-1.5 | Vicuna-7B | 78.5 | 62.0 | 50.0 | 66.8 | 58.2 | 85.9 | 1510.7 | 64.3 | 58.3 | 58.6 | 63.4 | 30.5 |
| LLaVA-1.5+Ours | Vicuna-7B | **80.3** | **63.7** | **51.9** | **69.1** | **60.8** | **87.7** | **1536.2** | **66.8** | **60.5** | **60.2** | **65.6** | **32.4** |
| LLaVA-1.5 | Vicuna-13B | 80.0 | 63.3 | 53.6 | 71.6 | 61.3 | 85.9 | 1531.3 | 67.7 | 63.6 | 61.6 | 70.7 | 35.4 |
| LLaVA-1.5+Ours | Vicuna-13B | **82.3** | **65.7** | **55.2** | **73.7** | **63.9** | **88.7** | **1584.7** | **69.8** | **65.2** | **63.4** | **72.6** | **38.3** |
| LLaVA-Next | Vicuna-7B | 81.8 | 64.2 | 57.6 | 70.1 | 64.9 | 86.5 | 1519 | 67.4 | 60.6 | 70.2 | 81.6 | 43.9 |
| LLaVA-Next+Ours | Vicuna-7B | **82.9** | **65.4** | **58.9** | **70.8** | **65.1** | **88.9** | **1551** | **69.9** | **62.5** | **71.0** | **82.9** | **44.8** |

Table 3: **Comparative analysis of fine-tuning spaces and flexibility levels** on ScienceQA test set [42]. All experiments utilize MemVP [26] with LLaMA-13B as the backbone language model. The notation is defined as follows: $\mathbf{W}_s^D$ represents diagonal scaling matrices, $\mathbf{W}_s^{B-D}$ denotes block-diagonal scaling matrices. $\mathbf{W}_s^B$ indicates banded scaling matrices, and $\mathbf{W}_{se}^*$ corresponds to Euclidean space fine-tuning matrices. Key parameters include $d$ for banded size and $\frac{n}{r}$ for block size. $\otimes_c$: Möbius matrix multiplication.

| Method | #Trainable Params (M) | $d$ | $\frac{n}{r}$ | $\otimes_c$ | Average |
|---|---|---|---|---|---|
| MemVP | 5.5 | - | - | - | 93.78 |
| *Efficient training* | | | | | |
| +$\mathbf{W}_{se}^D$ | 5.55 (+0.05) | 0 | 1 | - | 93.81 (+0.03) |
| +$\mathbf{W}_{se}^{B-D}$ | 5.64 (+0.14) | - | 2 | - | 93.70 (−0.08) |
| +$\mathbf{W}_{se}^B$ | 5.71 (+0.21) | 1 | - | - | 93.65 (−0.13) |
| *Efficient training in hyperbolic space* | | | | | |
| +$\mathbf{W}_s^D$ | 5.55 (+0.05) | 0 | 1 | ✗ | 93.91 |
| +$\mathbf{W}_s^D$ | 5.55 (+0.05) | 0 | 1 | ✓ | 94.72 (+0.94) |
| +$\mathbf{W}_s^{B-D}$ | 5.64 (+0.14) | - | 2 | ✓ | 94.79 (+1.01) |
| | 5.78 (+0.28) | - | 4 | ✓ | 94.84 |
| | 6.08 (+0.58) | - | 8 | ✓ | 94.82 |
| +$\mathbf{W}_s^B$ | 5.71 (+0.21) | 1 | - | ✓ | **94.89 (+1.11)** |
| | 5.86 (+0.36) | 2 | - | ✓ | 94.82 |
| | 6.15 (+0.65) | 4 | - | ✓ | 94.83 |

Table 4: **Ablation studies of HyperET across vision encoders with varying granularity levels** on ScienceQA test set.

| Method | Lang. Model | Vision Encoder | Average |
|---|---|---|---|
| MemVP | LLaMA-13B | DINOV2 | 91.47 |
| MemVP | LLaMA-13B | SAM | 91.16 |
| *Efficient training* | | | |
| +$\mathbf{W}_{se}^D$ | LLaMA-13B | DINOV2 | 91.98 (+0.51) |
| +$\mathbf{W}_{se}^D$ | LLaMA-13B | SAM | 92.05 (+0.89) |
| *Efficient training in hyperbolic space* | | | |
| +$\mathbf{W}_s^D$ | LLaMA-13B | DINOV2 | 93.38 (+1.91) |
| +$\mathbf{W}_s^D$ | LLaMA-13B | SAM | 93.74 (+2.58) |

Table 5: **Ablation study on the key components of HyperET** on selected five MLLM benchmarks. We here realize the extra learnable parameters as full matrices, i.e., $\mathbf{W}_s$. $\otimes_c$: Möbius matrix multiplication. $\mathbf{W}_{se}$ corresponds to Euclidean space fine-tuning matrices with the same number of parameters.

| Method | VQAv2 | GQA | VW | SQA | TVQA |
|---|---|---|---|---|---|
| Baseline | 80.0 | 63.3 | 53.6 | 71.6 | 61.3 |
| *Efficient training* | | | | | |
| +$\mathbf{W}_{se}$ | 80.8 | 63.8 | 53.8 | 71.7 | 61.8 |
| *Efficient training in hyperbolic Space* | | | | | |
| +$\mathbf{W}_s$ | **82.3** | **65.7** | **55.2** | **73.7** | **63.9** |
| −$\otimes_c$ | 81.1 | 64.0 | 53.9 | 71.9 | 62.1 |

## 5.3 Ablation Study on Fine-tuning

Here, we do an ablation study on the ScienceQA [42], employing MemVP [26] as the backbone.

**Introducing Different Flexibility into HyperET.** We systematically investigate our proposed three distinct parameterization strategies for HyperET 's learnable components. These strategies include diagonal scaling matrices $\mathbf{W}_s^D$ and their two extended variants: banded scaling matrices $\mathbf{W}_s^B$ and block-diagonal scaling matrices $\mathbf{W}_s^{B-D}$, which systematically increase parameter flexibility through progressively relaxed structural constraints. As evidenced in Table 3, enhanced flexibility—achieved through $\mathbf{W}_s^B$ fine-tuning—provides only marginal performance improvements (0.17% accuracy gain over $\mathbf{W}_s^D$) while introducing over-fitting risks when further increasing learnable parameters via expanded banded sizes. These findings demonstrate that our fine-tuning method effectively accomplishes two key objectives: (1) efficient adaptation to optimal hyperbolic radii and (2) robust cross-modal alignment for downstream tasks, all while maintaining parameter efficiency.

**Effectiveness of Training in Hyperbolic Space.** To isolate the performance gains attributable to hyperbolic space rather than parameter increases, we conduct controlled experiments by adding an equivalent number of parameters through matrix multiplication in Euclidean space, as shown in Table 3. The results demonstrate that such parameter augmentation either yields no significant performance improvement or even degrades model performance. This empirical evidence strongly validates the necessity of employing hyperbolic space for adjusting vision encoder in MLLMs, as the observed improvements in Table 3 cannot be explained by mere parameter increases.

**Necessity of Mobiüs matrix multiplication.** We conduct ablation studies to evaluate the impact of the Möbius matrix multiplication operation. As shown in Table 3, a comparison between rows 6 and 7 reveals that standard matrix multiplication (denoted by" ✗") underperforms 0.81% compared to the Möbius matrix multiplication, (represented by "✓"). This performance gap underscores the essential role of Möbius operations in precisely adjusting the hyperbolic radius of visual representations.

**Discussion of Different Vision Encoder.** To further clearly demonstrate the primary contributing factor behind the performance improvements, we present additional experimental results designed to isolate and eliminate the influence of merely increasing the number of trainable parameters. As shown in Table 4, fine-tuning SAM [29] and DINOv2 [46] in Euclidean space with the same number of additional parameters as HyperET yields only marginal gains. In contrast, fine-tuning with HyperET results in substantial performance improvements. These findings clearly illustrate that our main contribution arises specifically from the proposed hyperbolic radius adjustment mechanism, rather than simply from introducing more learnable parameters.

## 5.4 Ablation Study on Pre-training

To further assess the contribution of each key component in the proposed HyperET, we isolate each component in separate experiments, and the results are presented in Table 5. Comparing row 2 and row 3, training in Euclidean space instead of hyperbolic space leads to significantly lower performance, indicating that the granularity gap mismatch cannot be effectively addressed in Euclidean space. The observed improvement over the baseline is primarily attributable to the increased number of parameters rather than the alignment of granularity levels. Intriguingly, comparing row 2 and row 4, simply transitioning from Euclidean space to hyperbolic space yields a slight improvement, which we attribute to the inherent ability of hyperbolic space to capture granularity levels. However, this strategy lacks explicit mechanisms for hyperbolic radius adjustment, limiting its effectiveness in addressing granularity mismatches. Finally, by adjusting the hyperbolic radius through Möbius

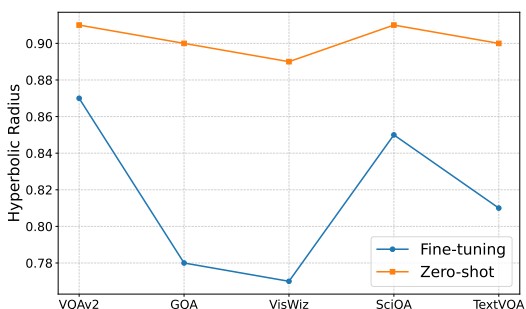

Figure 2: **Visualization of hyperbolic radius changes in visual representation after training** across different MLLM benchmarks. Normalizing the hyperbolic radius to a range of 0–1 facilitates comparison. A smaller hyperbolic radius corresponds to a more low granularity level of visual representation. "Zero-shot": maintaining the pretrained weights of the vision encoder, i.e., CLIP, without additional training.

matrix multiplication and integrating it with learnable matrices $\mathbf{W}_s$, the results show significant performance improvements, e.g., a 2.1% gain on the TextVQA [53], demonstrating the effectiveness of HyperET. Additionally, we visualize the changes in the hyperbolic radius in Fig. 2 after training, and the observed change in the hyperbolic radius indicates the MLLM's requirement for multi-granularity levels, demonstrating the necessity of our proposed HyperET.

## 6 Conclusion

This work proposes an efficient training paradigm (HyperET) for multi-modal large language models in hyperbolic space. By dynamically adjusting the hyperbolic radius of visual representations through learnable matrices and Möbius multiplication operations, HyperET effectively bridges the granularity gap between visual and textual modalities at an arbitrary granularity level. Our experiments across multiple MLLM benchmarks demonstrate that HyperET consistently improve existing pre-training and fine-tuning baselines by large margins with less than 1% additional parameters.

**Acknowledgment.** This work was supported by the NSFC under Grant 62322604, 62176159, and in part by the Shanghai Municipal Science and Technology Major Project under Grant 2021SHZDZX0102.

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

# Appendix of HyperET

This appendix provides additional theoretical and empirical details to support our work. Section A introduces the formal definition of hyperbolicity. Section B presents the full derivation of the core theorem introduced in the main manuscript. Section D offers an extended empirical comparison between HyperET and representative training methods in hyperbolic space, i.e., MERU, further validating the effectiveness of HyperET.

## A  The Definition of Hyperbolicity

This section introduces the concepts of hyperbolicity utilized in this work, including Möbius addition operation, Möbius multiplication operation, and the tangent space.

**Möbius addition operation.** For $\mathbf{X}, \mathbf{Y} \in \mathbb{D}_c^n$, the Möbius addition operation is defined as:

$$\mathbf{X} \oplus_c \mathbf{Y} := \frac{(1 + 2c\langle \mathbf{X}, \mathbf{Y} \rangle + c\|\mathbf{Y}\|^2)\mathbf{X} + (1 - c\|\mathbf{X}\|^2)\mathbf{Y}}{1 + 2c\langle \mathbf{X}, \mathbf{Y} \rangle + c^2\|\mathbf{X}\|^2\|\mathbf{Y}\|^2}. \tag{14}$$

**Möbius scalar multiplication operation.** For a scalar $r \in \mathbb{R}$ and a vector $\mathbf{X} \in \mathbb{D}_c^n$, the Möbius scalar multiplication operation is defined as:

$$r \otimes_c \mathbf{X} := (1/\sqrt{c})\tanh(r \cdot \tanh^{-1}(\sqrt{c}\|\mathbf{X}\|))\frac{\mathbf{X}}{\|\mathbf{X}\|}. \tag{15}$$

**Möbius matrix multiplication operation.** Refer to the relation definition in [28], for a matrix $\mathbf{M} \in \mathbb{R}^{n \times n}$ and and a point $\mathbf{X} \in \mathbb{D}_c^n$, if $\mathbf{MX} \neq \mathbf{0}$, the Möbius matrix multiplication operation is defined as:

$$\mathbf{M} \otimes_c \mathbf{X} := (\frac{1}{\sqrt{c}})\tanh\left(\frac{\|\mathbf{MX}\|}{\|\mathbf{X}\|} \cdot \tanh^{-1}(\sqrt{c}\|\mathbf{X}\|)\right)\frac{\mathbf{MX}}{\|\mathbf{MX}\|}.$$

**Tangent Space.** The tangent space $\mathcal{T}_{\mathbf{X}}^c\mathbb{D}_c^n$ at a point $\mathbf{X} \in \mathbb{D}_c^n$ is the first order approximation of $\mathbb{D}_c^n$, which is an $n$-dimensional Euclidean space. The tangent space $\mathcal{T}_{\mathbf{X}}^c\mathbb{D}_c^n$ and $\mathbb{D}_c^n$ are mapped to each other by exponential $(\mathcal{T}_{\mathbf{x}}^c\mathbb{D}_c^n \mapsto \mathbb{D}_c^n : \exp_{\mathbf{X}}^{\mathbb{D},c}(\cdot))$ and logarithmic $(\mathbb{D}_c^n \mapsto \mathcal{T}_{\mathbf{X}}^c\mathbb{D}_c^n : \log_{\mathbf{X}}^{\mathbb{D},c}(\cdot))$ maps, respectively. For any $\mathbf{X}, \mathbf{Y} \in \mathbb{D}_c^n$ and $\mathbf{V} \in \mathcal{T}_{\mathbf{X}}^c\mathbb{D}_c^n$, the mapping functions are given for $\mathbf{V} \neq \mathbf{0}$ and $\mathbf{Y} \neq \mathbf{X}$ by:

$$\exp_{\mathbf{X}}^{\mathbb{D},c}(\mathbf{V}) = \mathbf{X} \oplus_c \left(\tanh\left(\sqrt{c}\frac{\lambda_{c,\mathbf{x}}\|\mathbf{V}\|}{2}\right)\frac{\mathbf{V}}{\sqrt{c}\|\mathbf{V}\|}\right), \tag{16}$$

$$\log_{\mathbf{X}}^{\mathbb{D},c}(\mathbf{Y}) = \frac{2}{\sqrt{c}\lambda_{c,\mathbf{X}}}\tanh^{-1}\left(\sqrt{c}\|-\mathbf{X} \oplus_c \mathbf{Y}\|\right)\frac{-\mathbf{X} \oplus_c \mathbf{Y}}{\|-\mathbf{X} \oplus_c \mathbf{Y}\|}. \tag{17}$$

## B  Derivation of the Theorem

**Theorem 1** (Hyperbolic Radius Scaling) For a point $\mathbf{X} \in \mathbb{D}_c^n$ in hyperbolic space, the hyperbolic radius adjustment function is expanded as follows:

$$\begin{aligned} s \cdot \text{Rad}_{\mathbf{X}} &= \frac{2}{\sqrt{c}}(s\frac{\sqrt{c}}{2}\text{Rad}_{\mathbf{X}}) \\ &= \frac{2}{\sqrt{c}}\tanh^{-1}(\sqrt{c}\|s \otimes_c \mathbf{X}\|) \\ &= \text{Rad}_{s \otimes_c \mathbf{X}}. \end{aligned} \tag{18}$$

where $\otimes_c$ here is instantiated as a Mobiüs scalar multiplication operation. Therefore, hyperbolic radius adjustment can be precisely controlled through $\otimes_c$ between $s$ and $\mathbf{X}$, with the scaling coefficient $s$ serving as a primary learnable parameter.

*Proof.* In a hyperbolic space, considering a point $\mathbf{X} \in \mathbb{D}_c^n$ with hyperbolic radius $\text{Rad}_{\mathbf{X}} := (2/\sqrt{c})\tanh^{-1}(\sqrt{c}\|\mathbf{X}\|)$, the detailed expansion of the hyperbolic radius adjustment function is

formulated as:

$$
\begin{aligned}
s \cdot \mathrm{Rad}_{\mathbf{X}} &= \frac{2}{\sqrt{c}}(s\frac{\sqrt{c}}{2}\mathrm{Rad}_{\mathbf{X}}) \\
&= \frac{2}{\sqrt{c}}\tanh^{-1}(\tanh(s\frac{\sqrt{c}}{2}\mathrm{Rad}_{\mathbf{X}})) \\
&= \frac{2}{\sqrt{c}}\tanh^{-1}(\frac{\sqrt{c}}{\sqrt{c}}\tanh(s\frac{\sqrt{c}}{2}\mathrm{Rad}_{\mathbf{X}})\frac{\|\mathbf{X}\|}{\|\mathbf{X}\|}) \\
&= \frac{2}{\sqrt{c}}\tanh^{-1}(\sqrt{c}\left\|\frac{\tanh(s\frac{\sqrt{c}}{2}\mathrm{Rad}_{\mathbf{X}})}{\sqrt{c}\|\mathbf{X}\|}\mathbf{X}\right\|) \\
&= \frac{2}{\sqrt{c}}\tanh^{-1}(\sqrt{c}\left\|\frac{\tanh(s\tanh^{-1}(\sqrt{c}\|\mathbf{X}\|))}{\sqrt{c}\|\mathbf{X}\|}\mathbf{X}\right\|) \\
&= \frac{2}{\sqrt{c}}\tanh^{-1}(\sqrt{c}\|s\otimes_c \mathbf{X}\|) \\
&= \mathrm{Rad}_{s\otimes_c\mathbf{X}}.
\end{aligned} \tag{19}
$$

Consequently, according to Eq. (19), the scaling scalar $s$ can adjust the hyperbolic radius $\mathrm{Rad}_{\mathbf{X}}$ via the Möbius scalar multiplication operation. ∎

**Theorem 2** (Hyperbolic Radius Flexibility Scaling) For a point $\mathbf{X} \in \mathbb{D}_c^n$ in hyperbolic space and a scaling matrix $\mathbf{X}_s \in \mathbb{R}^{n\times n}$, the flexible radius adjustment function is formally defined through Eq. (7) and Mobiüs matrix multiplication operations as follows:

$$
\begin{aligned}
\frac{\|\mathbf{X}_s\mathbf{X}\|}{\|\mathbf{X}\|}\cdot\mathrm{Rad}_{\mathbf{X}} &= \frac{2}{\sqrt{c}}(\frac{\|\mathbf{X}_s\mathbf{X}\|}{\|\mathbf{X}\|}\frac{\sqrt{c}}{2}\mathrm{Rad}_{\mathbf{X}}) \\
&= \frac{2}{\sqrt{c}}\tanh^{-1}(\sqrt{c}\|\mathbf{X}_s\otimes_c\mathbf{X}\|) \\
&= \mathrm{Rad}_{\mathbf{X}_s\otimes_c\mathbf{X}}.
\end{aligned} \tag{20}
$$

*Proof.* In a hyperbolic space, considering a point $\mathbf{X}\in\mathbb{D}_c^n$ and a scaling matrix $\mathbf{X}_s\in\mathbb{R}^{n\times n}$, the detailed expansion of the hyperbolic radius flexibility adjustment function can be formulated as:

$$
\begin{aligned}
&\frac{\|\mathbf{X}_s\mathbf{X}\|}{\|\mathbf{X}\|}\cdot\mathrm{Rad}_{\mathbf{X}} \\
&= \frac{2}{\sqrt{c}}\tanh^{-1}(\tanh(\frac{\|\mathbf{X}_s\mathbf{X}\|}{\|\mathbf{X}\|}\frac{\sqrt{c}}{2}\mathrm{Rad}_{\mathbf{X}})\frac{\|\mathbf{X}_s\mathbf{X}\|}{\|\mathbf{X}_s\mathbf{X}\|}) \\
&= \frac{2}{\sqrt{c}}\tanh^{-1}\left(\left\|\frac{\tanh(\frac{\|\mathbf{X}_s\mathbf{X}\|}{\|\mathbf{X}\|}\frac{\sqrt{c}}{2}\mathrm{Rad}_{\mathbf{X}})}{\|\mathbf{X}_s\mathbf{X}\|}\mathbf{X}_s\mathbf{X}\right\|\right) \\
&= \frac{2}{\sqrt{c}}\tanh^{-1}\left(\sqrt{c}\left\|\frac{\tanh(\frac{\|\mathbf{X}_s\mathbf{X}\|}{\|\mathbf{X}\|}\frac{\sqrt{c}}{2}\mathrm{Rad}_{\mathbf{X}})}{\sqrt{c}\|\mathbf{X}_s\mathbf{X}\|}\mathbf{X}_s\mathbf{X}\right\|\right) \\
&= \frac{2}{\sqrt{c}}\tanh^{-1}\left(\sqrt{c}\left\|(1/\sqrt{c})\tanh\left(\frac{\|\mathbf{X}_s\mathbf{X}\|}{\|\mathbf{X}\|}\tanh^{-1}\left(\sqrt{c}\|\mathbf{X}\|\right)\right)\frac{\mathbf{X}_s\mathbf{X}}{\|\mathbf{X}_s\mathbf{X}\|}\right\|\right) \\
&= \frac{2}{\sqrt{c}}\tanh^{-1}(\sqrt{c}\|\mathbf{X}_s\otimes_c\mathbf{X}\|) \\
&= \mathrm{Rad}_{\mathbf{X}_s\otimes_c\mathbf{X}}.
\end{aligned} \tag{21}
$$

Consequently, according to Eq. (21), the scaling matrix $\mathbf{X}_s$ can adjust the hyperbolic radius $\mathrm{Rad}_{\mathbf{X}}$ via the Möbius matrix multiplication operation. ∎

# C  Validating HyperET via Comparison with Hyperbolic Training Methods

To further validate the effectiveness of our approach, we also compare HyperET with representative training methods that utilize hyperbolic space, i.e., MERU [17]. MERU [17] and its variants establish an asymmetric visual-semantic hierarchy that emphasizes a subordinate relationship wherein textual features dominate visual ones, inherently intensifying the granularity gap between text and visual modalites, leading to inferior performance. In contrast, HyperET attempts to align visual and textual modalities at arbitrary granularity levels. As shown in Table 6, our experimental results demonstrate that HyperET achieves superior performance compared to MERU, empirically validating the effectiveness and contribution of HyperET in the context of hyperbolic geometry.

Table 6: **Comparison with MERU [17]** on 12 MLLM benchmarks, including VQAv2 [21], GQA [25], VW: VisWiZ [22], SQA: ScienceQA-IMG [42], TVQA: TextVQA [53], PE: POPE [37], ME: MME [67], MB: MMBench [41], MB$^{CN}$: MMBench-Chinese [41], SD: SEED-Bench [34], LVA$^W$: LLaVA-Bench (In-the-Wild) [40] and M-Vet [68]. Lan. Model: Language model. Benchmark names are abbreviated to consistent with the main manuscript. "Ours": we here realize the extra learnable parameters as full matrices, i.e., $\mathbf{W}_s$. Vision encoder: CLIP.

| Method | Lan. Model | VQAv2 | GQA | VW | SQA | TVQA | PE | ME | MB | MB$^{CN}$ | SD | LVA$^W$ | M-Vet |
|---|---|---|---|---|---|---|---|---|---|---|---|---|---|
| LLaVA-1.5 | Vicuna-13B | 80.0 | 63.3 | 53.6 | 71.6 | 61.3 | 85.9 | 1531.3 | 67.7 | 63.6 | 61.6 | 70.7 | 35.4 |
| LLaVA-1.5+MERU | Vicuna-13B | 78.9 | 61.0 | 48.0 | 65.7 | 60.2 | 84.7 | 1456.8 | 62.3 | 59.1 | 56.6 | 68.4 | 32.5 |
| LLaVA-1.5+Ours | Vicuna-13B | **82.3** | **65.7** | **55.2** | **73.7** | **63.9** | **88.7** | **1584.7** | **69.8** | **65.2** | **63.4** | **72.6** | **38.3** |

# D  Validation on Non-Transformer MLLMs

To demonstrating our method's effectiveness on different architectures, we experiment HyperET on a non-Transformer architecture [72], i.e., Mamba. As shown in Table 7, our method yields consistent gains across all MLLM benchmarks, further demonstrating the robustness and generalizability of our approach.

Table 7: **Comparison with Cobra [72]** on 5 MLLM benchmarks, including VQAv2 [21], GQA [25], VW: VisWiZ [22], TVQA: TextVQA [53], PE: POPE [37]. Benchmark names are abbreviated to consistent with the main manuscript. "Ours": we here realize the extra learnable parameters as full matrices, i.e., $\mathbf{W}_s$.

| Method | VQAv2 | GQA | VW | TVQA | PE |
|---|---|---|---|---|---|
| Cobra | 79.2 | 63.9 | 56.2 | 59.5 | 87.6 |
| Cobra+Ours | **80.9** | **65.2** | **56.9** | **60.6** | **88.8** |

