# OpenReview forum: "HyperET: Efficient Training in Hyperbolic Space for Multi-modal Large Language Models"
_NeurIPS.cc/2025/Conference — NeurIPS 2025 oral_

### Official Review · Reviewer_g5HU · 2025-06-30

**Clarity:** 3
**Significance:** 3
**Originality:** 3
**Rating:** 4
**Confidence:** 4

**Summary:**

The paper proposes HyperET, a training paradigm for multi-modal large language models that operates in hyperbolic space to address purported granularity mismatches between visual and textual modalities. The authors claim that existing vision encoders like CLIP and SAM are inefficient because they align with language at single granularity levels, and propose using hyperbolic radius adjustment through Möbius multiplication operations to enable alignment at arbitrary granularity levels. The method demonstrates improvements across MLLM benchmarks with less than 1% additional parameters.

**Questions:**

- How do you control for the possibility that improvements stem simply from additional learnable parameters rather than the specific hyperbolic operations?
- How does the method perform when the "optimal" granularity level varies within a single input (eg images with both fine-grained and coarse-grained elements)?
- How sensitive is the method to the choice of hyperbolic curvature parameter c?
- What happens to the hyperbolic structure during backpropagation, and how do gradient flows behave in the constrained hyperbolic manifold?
- What is the theoretical capacity of the hyperbolic space compared to Euclidean representations. Could similar benefits be achieved through other geometric approaches?
- Can you offer a more detailed explanation for the observed radius changes in fig2?
- The learnable matrices seem to be applied in the attention layer. Is this choice based on empirical testing? How would applying the HyperET module to other components (such as the MLP projector) affect performance?

**Ethical Concerns:**

["NO or VERY MINOR ethics concerns only"]

**Final Justification:**

Most of my questions have been resolved.

**Limitations:**

yes

**Quality:**

3

**Strengths And Weaknesses:**

pros

- The core idea of leveraging hyperbolic geometry to explicitly model and adjust the semantic granularity seems interesting
- The method is efficient (with 1% additional parameters), a practical consideration.
- The paper is well written.

cons

- see the questions below

---

> ### Author Rebuttal · Authors · 2025-07-31
>
> **Q1: How do you control for the possibility that improvements stem simply from additional learnable parameters rather than the specific hyperbolic operations?**
>
> **A1**: We have presented experimental results to isolate and eliminate the influence of merely increasing the number of trainable parameters. As shown in Table 5 in the main manuscript, fine-tuning MLLMs in Euclidean space with the same number of additional parameters as HyperET yields only marginal gains. In contrast, fine-tuning with HyperET results in substantial performance improvements. These findings clearly illustrate that our main contribution arises specifically from the proposed hyperbolic radius adjustment mechanism, rather than simply from introducing more learnable parameters.
>
> **Q2: How does the method perform when the "optimal" granularity level varies within a single input (eg images with both fine-grained and coarse-grained elements)?**
>
> **A2**: There might be a misunderstanding here. The term "granularity'' refers to an intrinsic property of a task, e.g. semantic segmentation is a more granular task than image classification, rather than elements in a single image. A granularity level can correspond to a range of hyperbolic radii. Thus, we assume the reviewer was asking how our method performs when the ``optimal'' hyperbolic radius varies within a single input. This is straightforward, since our method performs hyperbolic feature learning on the dense feature maps. The features at different locations naturally have varied hyperbolic radii. But, note that the variance of these hyperbolic radii is small (typically 0.01 in our experiments), evidencing that they belong to one hyperbolic radius range, corresponding to one granularity level.
>
> **Q3: How sensitive is the method to the choice of hyperbolic curvature parameter c?**
>
> **A3**: The curvature hyperparameter c is set to 0.01 in our experiments, but as shown in the table below, our method exhibits only minor performance variations when c ranges from 0.001 to 1. This demonstrates the stability of our approach across different curvature settings.
>
> | Curvature hyperparameter c | 1    | 0.1  | 0.01 | 0.001 |
> |----------------------------|------|------|------|-------|
> | Average (%)                | 93.56| 93.64| 93.78| 93.71 |
>
> *Table R1: Ablation study of curvature hyperparameter $c$ on ScienceQA Test Set.*
>
> **Q4: What happens to the hyperbolic structure during backpropagation, and how do gradient flows behave in the constrained hyperbolic manifold?**
>
> **A4**: Our method preserves the intrinsic hyperbolic geometry by learning to adjust only the radii of points within hyperbolic space. Gradient flows $\nabla_{\mathbb{H}}$ follow standard Euclidean backpropagation $\nabla_{\mathbb{E}}$, with an additional radius-aware scaling:
> $\nabla_{\mathbb{H}} = (1-\tanh^2(r)) \cdot \nabla_{\mathbb{E}}$, where $r$ is the radius of a point. This technique guarantees stability near the boundary of the Poincaré ball, ensuring updated points remain within the manifold. Furthermore, it is compatible with standard optimizers, e.g., Adam.
>
> **Q5: What is the theoretical capacity of the hyperbolic space compared to Euclidean representations. Could similar benefits be achieved through other geometric approaches?**
>
> **A5**: Hyperbolic space preserves a larger theoretical capacity than Euclidean space due to its intrinsic exponential volume growth, where the circumference scales as $C(r) \propto e^{r}$ compared to the linear $C(r) \propto r$ scaling in Euclidean space, where $r$ is radius. Other geometric approaches, e.g., kernel-based methods, lack clear geometric interpretability and often require manual kernel design. Hyperspherical space offers geometric interpretability but is hindered by intrinsic limitations, particularly its saturated volume growth, which restricts hierarchical granularity representation.
>
> **Q6: Can you offer a more detailed explanation for the observed radius changes in fig2?**
>
> **A6**: Yes. Figure 2 illustrates the variation of representation norms (hyperbolic radii) among different tasks. More concretely, the VQAv2 dataset is a general visual question answering benchmark that typically requires representations with larger norms, whereas the GQA dataset emphasizes detailed spatial reasoning tasks, leading to representations characterized by comparatively smaller norms.
>
> **Q7: The learnable matrices seem to be applied in the attention layer. Is this choice based on empirical testing? How would applying the HyperET module to other components (such as the MLP projector) affect performance?**
>
> **A7**: The placement of learnable matrices within the attention layers is consistent with mainstream parameter-efficient methods (e.g., LoRA) and is empirically validated, as shown in the table below. The observed performance gains may result from a hypothesis that interactions in the attention layers more effectively influence the adjustment of the hyperbolic radius.
>
> | Module               | Average |
> |----------------------|---------|
> | Attention layer (Ours) | **93.78** |
> | MLP Projector        | 93.11   |
>
> *Table R2: Ablation study of placement of learnable matrices on ScienceQA Test Set.*

---

> > ### Comment · Reviewer_g5HU · 2025-08-04
> >
> > Thanks for your clarifications. Most of my questions have been resolved. I will continue to follow the discussions and feedback from other reviewers to decide my final score.

---

> > > ### Author Response · Authors · 2025-08-04
> > >
> > > Dear Reviewer g5HU,
> > >
> > > We sincerely thank you for your valuable and constructive reviews, and also sincerely thank you for your feedback.
> > >
> > > Best Regards,
> > >
> > > Authors.

---

### Official Review · Reviewer_hmUq · 2025-07-02

**Clarity:** 2
**Significance:** 2
**Originality:** 3
**Rating:** 4
**Confidence:** 4

**Summary:**

This paper identifies a "granularity gap" between vision and language modalities as a key source of inefficiency in training Multi-modal Large Language Models (MLLMs). It proposes HyperET, a novel training paradigm that uses hyperbolic space to model and align these different levels of granularity. The core idea is to use the "hyperbolic radius" as a proxy for visual granularity, which is then dynamically adjusted using learnable, parameter-efficient scaling matrices and Möbius multiplication operations. By adding less than 1% of new parameters, HyperET is shown to consistently improve the performance of existing MLLMs in both fine-tuning and pre-training scenarios across a wide range of benchmarks.

**Questions:**

1. Can you provide a qualitative analysis to make the concept of "granularity adjustment" more concrete? For instance, visualizing attention maps before and after applying HyperET could show if the model’s focus shifts from whole objects to finer details as the hyperbolic radius changes.

2. To better establish the generality of your method, could you more directly compare HyperET's performance boost on a standard CLIP encoder versus an encoder that is already considered more granular, like DINOv2? This would clarify if the benefit is primarily in "fixing" CLIP or if it's a more universal alignment tool.

3. The hyperbolic curvature -c is a key parameter in the methodology. Could authors discuss how this was chosen and how sensitive the model's performance is to this choice? This would improve the paper's practical guidance and reproducibility

**Ethical Concerns:**

["NO or VERY MINOR ethics concerns only"]

**Final Justification:**

Based on the rebuttal and the discussion with other reviewers, my concerns have been largely addressed. The clarifications provided were helpful, and as a result, I'm raising my score from 3 to 4.

**Quality:**

2

**Strengths And Weaknesses:**

## Strengths

1. The paper's framing of MLLM inefficiency as a "granularity gap" is insightful, and using hyperbolic geometry as a solution is a principled and elegant approach.
2. The method is rigorously tested across multiple models and tasks. Crucially, strong ablation studies confirm that the performance gains stem directly from the proposed hyperbolic mechanisms, not just from adding parameters.
3. HyperET achieves significant performance improvements while adding negligible computational overhead (<1% additional parameters), making it a highly practical and accessible method.

## Weaknesses
1. The paper does not provide a qualitative analysis of why the model adjusts the hyperbolic radius for different tasks, missing an opportunity to offer deeper intuition behind the "granularity adjustment".
2. The reliance on advanced concepts from hyperbolic geometry adds significant complexity that may not always be justified by the margin of performance improvement.
3. The theory suggests that granularity is adjusted dynamically per input, a powerful feature that is not explicitly analyzed or discussed in the experimental results.

---

> ### Author Rebuttal · Authors · 2025-07-31
>
> **W1: The paper does not provide a qualitative analysis of why the model adjusts the hyperbolic radius for different tasks, missing an opportunity to offer deeper intuition behind the "granularity adjustment".**
>
> **A1**: Different tasks typically require distinct granularity levels (pixel-, object-, or image-level, etc), and vanilla CLIP suffers from granularity mismatch on various tasks that degrades performance. By adaptively adjusting hyperbolic radii of CLIP's feature embeddings, we align model granularity with task requirements. Experiments on open-vocabulary semantic segmentation and object detection demonstrate consistent and significant improvements when matching granularity, e.g., 12.6\% mIoU improvement on the PASCAL-Context dataset  (see the table below).
>
> | **Task/Feature granularity** | **Hyperbolic radius** | **Dataset** | **Performance** |
> |-----------------------------|-----------------------|-------------|-----------------|
> | *Open-Vocabulary Semantic Segmentation* | | | |
> | Pixel-level/Image-level | 7.5$\pm$ 0.03/8.2$\pm$ 0.05 | PC-459| 6.6 |
> | **Pixel-level/Pixel-level** | 6.1$\pm$ 0.03/7.2$\pm$ 0.05 | PC-459 | **19.2** |
> | *Open-Vocabulary Object Detection* | | | |
> | Object-level/Image-level | 7.5$\pm$ 0.03/8.2$\pm$ 0.05 | COCO | 36.4 / 21.6 |
> | **Object-level/Object-level** | 6.4$\pm$ 0.03/7.5$\pm$ 0.05 | COCO | **42.7 / 26.3** |
>
> *Table R1: Comparative performance across different granularity levels. For semantic segmentation: PASCAL-Context dataset (PC-459); for object detection: COCO dataset ($\text{AP}^{\text{Base}}_{50} / \text{\AP}^{\text{Novel}}_{50}$).*
>
> **W2: The reliance on advanced concepts from hyperbolic geometry adds significant complexity that may not always be justified by the margin of performance improvement.**
>
> **A2**: Our approach is not complex: it maps embeddings from Euclidean space into hyperbolic space through a mathematical transformation, adjusts their radii via scaling operations, and subsequently projects them back. Each step has been rigorously validated through ablation studies (see Tables 3 and 5 in the main manuscript), confirming the necessity and effectiveness of this lightweight geometric adaptation.
>
> **W3: The theory suggests that granularity is adjusted dynamically per input, a powerful feature that is not explicitly analyzed or discussed in the experimental results.**
>
> **A3**: Thank you for your suggestion. To demonstrate the capability of our method in dynamically adjusting granularity per input, we present a representative example from the 2nd iteration of the Visual Question Answering (VQAv2) dataset (Image ID: 53141, Question: "What is in the sky?"). In the image of this example, a flight is located at the center of the image against a clear sky background. The two below matrices show the hyperbolic radii of the features on the feature map extracted from the last layer of CLIP's image encoder, before and after the adjustment by our method, respectively. It can be observed that, our method adjusts the mean hyperbolic radius from 8.28 to 8.02,  shifting the granularity level from the original image-level to the fine-grained level, with smaller values concentrated on the object of interest. This effectively aligns feature granularity with the VQA task's requirements.
>
> **Before**
> |       |       |       |       |       |       |       |
> |-------|-------|-------|-------|-------|-------|-------|
> | 8.39  | 8.38  | 8.26  | 8.31  | 8.25  | 8.33  | 8.42  |
> | 8.24  | 8.27  | 8.29  | 8.36  | **8.23**  | **8.21**  | 8.34  |
> | **8.22**  | 8.30  | 8.28  | **8.22**  | 8.32  | 8.37  | 8.31  |
> | 8.35  | **8.21**  | **8.13**  | **8.11**  | **8.05**  | 8.36  | 8.36  |
> | 8.32  | 8.33  | **8.13**  | **8.09**  | **8.19**  | **8.21**  | 8.39  |
> | 8.41  | 8.32  | **8.23**  | 8.32  | 8.25  | **8.24**  | 8.25  |
> | 8.34  | 8.36  | 8.33  | 8.30  | 8.38  | 8.35  | 8.37  |
>
> **After**
>
> |       |       |       |       |       |       |       |
> |-------|-------|-------|-------|-------|-------|-------|
> | 8.02  | 8.19  | 8.06  | 8.11  | 8.07  | 8.12  | 8.02  |
> | 8.13  | 8.07  | 7.99  | 7.95  | 7.99  | 7.95  | 8.11  |
> | 8.07  | 8.00  | 7.99  | 7.89  | 7.92  | 7.98  | 8.14  |
> | 8.05  | 8.01  | 7.86  | **7.81**  | **7.89**  | 8.03  | 8.12  |
> | 8.02  | 8.03  | 7.93  | **7.87**  | 7.89  | 8.01  | 8.09  |
> | 8.21  | 8.02  | 7.99  | 8.02  | 8.05  | 8.03  | 8.05  |
> | 8.17  | 8.12  | 8.03  | 8.00  | 8.08  | 8.05  | 8.02  |
>
> **Q1:** **Can you provide a qualitative analysis to make the concept of "granularity adjustment" more concrete? For instance, visualizing attention maps before and after applying HyperET could show if the model's focus shifts from whole objects to finer details as the hyperbolic radius changes.**
>
> **A4:** We use the same example as in W3. The two matrices below represent the attention maps extracted from the last layer of CLIP's image encoder, before and after hyperbolic radius adjustment. These maps clearly demonstrate a shift in focus from coarse object-level regions to finer, more discriminative details. (1) Before adjustment, attention is broadly distributed across regions, e.g., the sky and background, with moderate activation values (e.g., 0.34-0.66) even in irrelevant areas. (2) After adjustment, the model sharply focuses on the central object (a flight), with high attention values (e.g., 0.75-0.91) in key regions, while background areas exhibit minimal activation (mostly < 0.1).
>
> **Before**
>
> |       |       |       |       |       |       |       |
> |-------|-------|-------|-------|-------|-------|-------|
> | 0.12  | 0.14  | 0.16  | 0.15  | 0.17  | 0.21  | 0.06  |
> | 0.25  | 0.27  | 0.21  | 0.03  | 0.25  | **0.31**  | 0.14  |
> | **0.34**  | 0.05  | **0.46**  | **0.35**  | **0.33**  | 0.12  | 0.21  |
> | 0.23  | **0.45**  | **0.56**  | **0.59**  | **0.43**  | 0.24  | 0.15  |
> | 0.12  | 0.24  | **0.66**  | **0.54**  | **0.53**  | **0.34**  | 0.16  |
> | 0.08  | 0.23  | **0.45**  | 0.23  | **0.43**  | 0.23  | **0.34**  |
> | 0.04  | 0.23  | **0.33**  | 0.26  | 0.04  | 0.15  | 0.09  |
>
> **After**
> |       |       |       |       |       |       |       |
> |-------|-------|-------|-------|-------|-------|-------|
> | 0.07  | 0.04  | 0.08  | 0.05  | 0.12  | 0.11  | 0.01  |
> | 0.09  | 0.07  | 0.11  | 0.01  | 0.04  | 0.01  | 0.06  |
> | 0.04  | 0.05  | 0.16  | 0.75  | 0.23  | 0.02  | 0.03  |
> | 0.03  | 0.05  | 0.76  | **0.91**  | **0.83**  | 0.14  | 0.05  |
> | 0.02  | 0.14  | 0.16  | 0.74  | 0.13  | 0.04  | 0.06  |
> | 0.01  | 0.12  | 0.15  | 0.03  | 0.03  | 0.13  | 0.04  |
> | 0.05  | 0.03  | 0.03  | 0.06  | 0.02  | 0.05  | 0.01  |
>
>
> **Q2:** **To better establish the generality of your method, could you more directly compare HyperET's performance boost on a standard CLIP encoder versus an encoder that is already considered more granular, like DINOv2? This would clarify if the benefit is primarily in "fixing" CLIP or if it's a more universal alignment tool.**
>
> **A5:** The reviewer may overlook some experimental results in our paper. We already provided such comparisons with non-CLIP-based MLLMs, which are presented in Tables 4 of the main manuscript, including employing DINOv2 and SAM as the vision encoder. In Table 4, our HyperET also yields consistent performance improvements. These results indicate that 1) the granularity mismatch also persists in more granular visual encoders, e.g., DINOv2 and SAM, as the required granularity levels vary among different tasks; 2) HyperET can explicitly address this issue, serving as a more universal alignment strategy to effectively unleash the full potential of these vision encoders.
>
> **Q3:** **The hyperbolic curvature -c is a key parameter in the methodology. Could authors discuss how this was chosen and how sensitive the model's performance is to this choice? This would improve the paper's practical guidance and reproducibility.**
>
> **A6:** The curvature hyperparameter c is set to 0.01 in our experiments. As shown in the table below, our method exhibits negligible performance variations when c ranges from 0.001 to 1. This demonstrates the our approach is not sensitive to the choice of the curvature hyperparameter c.
>
> | Curvature hyperparameter $c$ | 1 | 0.1 | 0.01 | 0.001 |
> |------------------------------|---|---|---|----|
> | Average (%) | 93.56 | 93.64 | 93.78 | 93.71 |
>
> *Table R2: Ablation study of curvature hyperparameter $c$ on ScienceQA Test Set.*

---

> ### Author Response · Authors · 2025-08-05
>
> Dear Reviewer hmUq,
>
> Thank you again for your constructive comments and valuable suggestions. As the author-reviewer discussion period is nearing its end, we wanted to briefly follow up to ensure our rebuttals have addressed your concerns.
> We would be very grateful for any further feedback.
>
> Best regards,
>
> The Authors

---

> > ### Comment · Reviewer_hmUq · 2025-08-06
> >
> > Thanks for the clarification, It resolves most of my concerns and I will raise the score

---

> > > ### Author Response · Authors · 2025-08-06
> > >
> > > Dear Reviewer hmUq:
> > >
> > > Thank you for your thoughtful review of our rebuttal and for reconsidering your evaluation. We appreciate your constructive feedback and will be sure to incorporate this analysis in our revised manuscript.
> > >
> > > Best regards,
> > >
> > > Authors.

---

### Official Review · Reviewer_1nn1 · 2025-07-03

**Clarity:** 2
**Significance:** 3
**Originality:** 2
**Rating:** 4
**Confidence:** 4

**Summary:**

This paper proposes an efficient training paradigm called HyperET to address the inefficiency of cross-modal alignment at multi-granularity levels in MLLMs. Existing MLLMs rely on vision encoders like CLIP and SAM, which only align with language at a single granularity level, leading to heavy computational demands. HyperET leverages the properties of hyperbolic space to dynamically adjust the hyperbolic radius of visual representations, enabling alignment between visual and textual modalities at arbitrary granularity levels. Specifically, it employs learnable matrices with Möbius multiplication operations (including diagonal, block-diagonal, and banded matrices), significantly improving the performance of existing MLLMs in both pre-training and fine-tuning tasks with less than 1% additional parameters

**Questions:**

1.	The ablation study shows that hyperbolic space training outperforms Euclidean space with the same number of parameters, but the mechanism behind this superiority is not fully explained. Why is hyperbolic space uniquely effective for granularity alignment, and are there cases where Euclidean space could be more suitable?
2.	The three matrix configurations (diagonal, block-diagonal, banded) are presented as flexible options, but their practical selection criteria are unclear. For specific tasks (e.g., high-resolution image understanding vs. low-level feature matching), how should researchers choose the optimal matrix type, and what is the theoretical basis for this choice?
3.	The paper claims that HyperET achieves significant improvements with less than 1% additional parameters, but it does not compare the computational overhead of hyperbolic space operations with standard Euclidean operations. Could you provide quantitative analysis on the training/inference speed and memory usage of HyperET compared to baseline methods?

**Ethical Concerns:**

["NO or VERY MINOR ethics concerns only"]

**Final Justification:**

Thanks for your response. It resolves most of my concerns, and I will maintain the score.

**Quality:**

3

**Strengths And Weaknesses:**

Strength：
1.	By adjusting the hyperbolic radius in hyperbolic space, it directly quantifies and aligns granular differences between vision and text, avoiding training inefficiencies caused by mismatched granularity in traditional methods
2.	Three parameter-efficient learnable matrix designs (diagonal, block-diagonal, banded) achieve significant performance gains with less than 1% additional parameters, reducing computational costs
3.	Based on hyperbolic geometry and Möbius operations, its effectiveness in granularity alignment is validated through theorems and derivations

Weakness：
1.	All experiments rely on CLIP as the vision encoder , leaving unanswered whether HyperET generalizes to other encoders or how it interacts with non-CLIP-based MLLMs.
2.	While three matrix configurations (diagonal, block-diagonal, banded) are proposed, the paper provides limited insight into when each is most effective. For example, it does not clarify why banded matrices with larger bandwidths do not consistently outperform simpler designs

---

> ### Author Rebuttal · Authors · 2025-07-31
>
> **W1: All experiments rely on CLIP as the vision encoder, leaving unanswered whether HyperET generalizes to other encoders or how it interacts with non-CLIP-based MLLMs.**
>
> **A1**: This is a misunderstanding. We already provided comparisons with non-CLIP-based MLLMs. These results are presented in Tables 4 of the main manuscript, including employing DINOv2 and SAM as the vision encoder. Our HyperET also yields consistent performance improvements using these visual encoders. These results indicate that 1) the granularity mismatch also persists in non-CLIP visual encoders, as the required granularity levels vary among different tasks; 2) the proposed HyperET can explicitly address this issue, effectively unleashing the full potential of these vision encoders.
>
> **W2: While three matrix configurations (diagonal, block-diagonal, banded) are proposed, the paper provides limited insight into when each is most effective. For example, it does not clarify why banded matrices with larger bandwidths do not consistently outperform simpler designs.**
>
> **A2**: In general, the banded configuration is preferable and typically outperforms diagonal and block-diagonal structures. However, for downstream tasks with limited data (e.g., the ScienceQA dataset), where fewer parameters are sufficient for adaptation, the block-diagonal configuration can achieve performance comparable to that of the banded strategy.
>
> **Q1: The ablation study shows that hyperbolic space training outperforms Euclidean space with the same number of parameters, but the mechanism behind this superiority is not fully explained. Why is hyperbolic space uniquely effective for granularity alignment, and are there cases where Euclidean space could be more suitable?**
>
> **A3**: Many prior studies [a,b,c] have demonstrated, both theoretically and empirically, that hyperbolic space is particularly effective for granularity alignment tasks, owing to its hierarchical geometry that inherently preserves multiple granularity levels, which is a fundamental advantage over Euclidean space. Nonetheless, for tasks involving flatter structures or single-level granularity (e.g., transferring CLIP to classification tasks), Euclidean space may still offer competitive performance theoretically.
>
> **Q2: The three matrix configurations (diagonal, block-diagonal, banded) are presented as flexible options, but their practical selection criteria are unclear. For specific tasks (e.g., high-resolution image understanding vs. low-level feature matching), how should researchers choose the optimal matrix type, and what is the theoretical basis for this choice?**
>
> **A4**: The practical criteria for selecting among diagonal, block-diagonal, and banded matrix structures primarily consider task complexity and data scale. (1) Diagonal matrices require the fewest parameters, making them suitable primarily for simple tasks (e.g., edge detection). (2) Block-diagonal matrices, which leverage intra-block parameter modeling, offer an effective balance between parameter efficiency and modeling capacity, making them particularly competitive in low-data scenarios (e.g., fine-tuning MLLMs on the ScienceQA dataset). (3) Banded matrices, which encompass block-diagonal structures and additionally incorporate local connections, generally achieve optimal performance for complex tasks with abundant data (e.g., high-resolution image understanding). In summary, the banded configuration is generally preferable; however, diagonal and block-diagonal structures remain competitive choices for simpler tasks or data-constrained scenarios.
>
> **Q3: The paper claims that HyperET achieves significant improvements with less than 1% additional parameters, but it does not compare the computational overhead of hyperbolic space operations with standard Euclidean operations. Could you provide quantitative analysis on the training/inference speed and memory usage of HyperET compared to baseline methods?**
>
> **A5**: Since we do not introduce any extra architectures, and the logarithmic and exponential maps can be completed within linear complexity [d], the training/inference overhead is negligible. --See table below
>
> | Method               | Training time (h) | Memory (GB) | FLOPs (T) | Latency (ms) |
> |----------------------|------------------|-------------|-----------|--------------|
> | LLaVA-1.5-7B         | 13.5             | 34.5        | 8.55      | 113.0        |
> | LLaVA-1.5-7B+Ours    | 13.6 (+0.1)      | 35.2 (+0.7) | 8.61 (+0.06) | 114.1 (+1.1) |
>
> *Table R1: Efficiency comparison with baseline methods.*
>
> [a] Poincaré Embeddings for Learning Hierarchical Representations. NeurIPS2017.
> [b] Hyperbolic Neural Networks. NeurIPS2018.
> [c] Hyperbolic Image Embeddings. CVPR2020.
> [d] Hyperbolic Fine-tuning for Large Language Models. ICML2024.

---

> > ### Author Response · Authors · 2025-08-07
> >
> > Dear Reviewer 1nn1,
> >
> > May we kindly ask if our responses have addressed your concerns? We look forward to further discussions and your feedback.
> >
> > Best Regards,
> >
> > Authors.

---

> > > ### Comment · Reviewer_1nn1 · 2025-08-07
> > >
> > > Thanks for your response. It resolves most of my concerns.

---

> > > > ### Author Response · Authors · 2025-08-07
> > > >
> > > > Dear Reviewer 1nn1,
> > > >
> > > > Thank you for your diligent review of my rebuttal and manuscript. I appreciate your valuable feedback and will incorporate your suggestions in the revised version.
> > > >
> > > > Should anything remain unclear, we are happy to provide further explanation.
> > > >
> > > > Sincerely,
> > > > Authors.

---

### Official Review · Reviewer_ijbW · 2025-07-03

**Clarity:** 3
**Significance:** 3
**Originality:** 3
**Rating:** 4
**Confidence:** 3

**Summary:**

This paper introduces HyperET, a novel and efficient training paradigm for multi-modal large language models (MLLMs) designed to address the "granularity mismatch" between visual and textual modalities. The core idea is to leverage the natural hierarchical structure of hyperbolic space to represent and align features at varying levels of granularity. By dynamically adjusting the hyperbolic radius of visual representations via a parameter-efficient learnable matrix and Möbius multiplication, HyperET enables visual features to align with their textual counterparts at an arbitrary granularity level. The authors conduct comprehensive experiments on both fine-tuning and pre-training scenarios, demonstrating that HyperET consistently improves the performance of state-of-the-art MLLMs (e.g., LLaVA, MemVP) with less than 1% additional parameters, notably reducing object hallucination. The method is well-motivated, theoretically sound, and empirically validated across multiple standard benchmarks.

**Questions:**

Please see the weaknesses. Does the proposed method's effectiveness extend to MLLMs with non-Transformer architectures, such as those based on Mamba?

**Ethical Concerns:**

["NO or VERY MINOR ethics concerns only"]

**Final Justification:**

Comprehensive experiments across different architectures are crucial to demonstrate the method's generalizability. The current version, lacking this, is not sufficient for a clear accept. Therefore, I will maintain my score of Borderline Accept.

**Limitations:**

Yes

**Quality:**

3

**Strengths And Weaknesses:**

### Strengths
1. Strong and Well-Defined Motivation: The paper effectively identifies a critical and practical problem in MLLM training: the granularity mismatch between vision encoders and language models. The argument that this mismatch leads to inefficiency and performance issues like hallucination is compelling and provides a solid foundation for the work.
2. Novel and Theoretically-Grounded Method: The proposed solution of using hyperbolic space to model and adjust semantic granularity is highly innovative and elegant. The connection between hyperbolic radius and granularity level is intuitive and provides a quantifiable framework for a previously abstract problem. The method is well-supported by clear mathematical proofs (Theorems 1 & 2), which adds significant credibility.
3. Comprehensive Experimental Validation: The empirical evaluation is a major strength.

### Weaknesses
1. Indirect Justification of the Core Assumption: The fundamental assumption—that hyperbolic radius directly corresponds to semantic granularity—is not empirically verified. The paper lacks qualitative or quantitative analysis to visually or otherwise demonstrate that features at different radii actually capture different levels of semantic detail (e.g., from pixels/parts to global concepts).
2. Insufficient Analysis of Computational Overhead: While the paper emphasizes "parameter efficiency," it does not discuss the computational efficiency in terms of training time or FLOPs. Operations in hyperbolic space are generally more complex than their Euclidean counterparts, and an analysis of the wall-clock time or computational overhead would provide a more complete picture of the method's "efficiency."
3. Lack of Hyperparameter Details: For example, the choice and sensitivity of the curvature hyperparameter c, which is crucial for the geometry of the hyperbolic space, are not discussed.

---

> ### Author Rebuttal · Authors · 2025-07-31
>
> **W1: Indirect Justification of the Core Assumption.**
>
> **A1**: Our core assumption that hyperbolic radius corresponds to semantic granularity is supported by both experimental results in prior works [a-c] and new empirical evidence. As shown in the Table R1 below, the hyperbolic radii of the features output from vanilla CLIP's text and image encoders are typically large (7.5 $\pm$ 0.03 and 8.2 $\pm$ 0.05, respectively), which are known to excel at capturing image-level semantics, but struggles with finer-grained understanding (e.g., only 6.6\% mIoU on the PASCAL-Context dataset and 36.4 mAP on base class object detection on the COCO dataset). After optimizing the hyperbolic radii of embeddings (reducing to 6.1 $\pm$ 0.03 and 7.2 $\pm$ 0.05 for segmentation and 6.4 $\pm$ 0.03 and 7.5 $\pm$ 0.05 for detection), we observe dramatic performance improvements across all benchmarks (e.g., a 12.6\% mIoU improvement on the PASCAL-Context dataset). These results demonstrate that features at different hyperbolic radii capture different levels of semantic detail. Moreover, the mean hyperbolic radii exhibit a progressive decrease from image-level (global concept, largest) to object-level (intermediate) and finally to pixel-level (smallest), demonstrating an intrinsic granularity hierarchy that adapts to different visual tasks. Another evidence is provided in Table R2 below. In this table, we calculate the hyperbolic radii of feature maps computed by three different visual encoders, i.e., CLIP, DINOv2 and SAM, on the ScienceQA dataset. Both DINOv2 and SAM are known as more fine-grained encoders than CLIP, where SAM is pre-trained on pixel-level tasks, endowing it with the finest granularity level. This granularity difference is precisely reflected in their hyperbolic radius values, where SAM, CLIP and DINOv2 exhibits the smallest (7.0$\pm$0.01), the largest (8.2$\pm$0.05) and the intermediate (7.4$\pm$0.09) hyperbolic radius, respectively. Moreover, DINOv2's broader granularity scope naturally results in greater variation in its hyperbolic radius range, consistent with its more diverse feature extraction capabilities.
>
> | Granularity level | Hyperbolic radius | Dataset | Performance |
> |------------------|------------------|---------|-------------|
> | *Open-Vocabulary Semantic Segmentation* | | | |
> | Image-level | 7.5$\pm$0.03 /8.2$\pm$0.05 | PC-459 | 6.6 |
> | **Pixel-level** | 6.1$\pm$0.03 /7.2$\pm$0.05 | PC-459 | **19.2** |
> | *Open-Vocabulary Object Detection* | | | |
> | Image-level | 7.5$\pm$0.03 /8.2$\pm$0.05 | COCO | 36.4 / 21.6 |
> | **Object-level** | 6.4$\pm$0.03 /7.5$\pm$0.05 | COCO | **42.7 / 26.3** |
>
> *Table R1: Comparative performance across different granularity levels. For semantic segmentation: PASCAL-Context dataset (PC-459); for object detection: COCO dataset ($\text{AP}^{\text{Base}}_{50} / \text{AP}^{\text{Novel}}_{50}$).*
>
> | Visual encoder | Hyperbolic radius |
> |---------------|------------------|
> | CLIP | 8.2$\pm$0.05 |
> | DINOv2 | 7.4$\pm$0.09 |
> | SAM | 7.1$\pm$0.01 |
>
> *Table R2: Comparison of hyperbolic radii across different visual encoders on the ScienceQA dataset.*
>
> **W2: Insufficient Analysis of Computational Overhead.**
>
> **A2**: Since we do not introduce any extra architectures, and the logarithmic and exponential maps can be completed within linear complexity [d], the training/inference overhead is negligible compared with the Euclidean counterpart. --See table below
>
> | Method | Training time (h) | Memory (GB) | FLOPs (T) | Latency (ms) |
> |--------|------------------|-------------|-----------|--------------|
> | LLaVA-1.5-7B | 13.5 | 34.5 | 8.55 | 113.0 |
> | LLaVA-1.5-7B+Ours | 13.6 (+0.1) | 35.2 (+0.7) | 8.61 (+0.06) | 114.1 (+1.1) |
>
> *Table R3: Efficiency comparison with baseline methods.*
>
> **W3: Lack of Hyperparameter Details.**
>
> **A3**:  The curvature hyperparameter c is set to 0.01 in our experiments. As shown in the table R4 below, our method exhibits negligible performance variations when c ranges from 0.001 to 1. This demonstrates our approach is not sensitive to the choice of the curvature hyperparameter c.
>
> | Curvature hyperparameter c | 1 | 0.1 | 0.01 | 0.001 |
> |---------------------------|------|------|------|-------|
> | Average (%) | 93.56 | 93.64 | 93.78 | 93.71 |
>
> *Table R4: Ablation study of curvature hyperparameter c on ScienceQA Test Set.*
>
> **Q1: Does the proposed method's effectiveness extend to MLLMs with non-Transformer architectures, such as those based on Mamba?**
>
> **A4**:  Thank you for your valuable suggestions. We have validated the robustness of our method using the mainstream architecture for MLLMs, i.e., Transformers. Although few recent studies have explored non-Transformer architectures, e.g., Mamba, they are proposed primarily for acceleration purposes, and most of their performance remains comparable to that of Transformer-based models. Since our approach directly adjusts the hyperbolic radius of features without relying on any specific architectural design, it is likely to be compatible with Mamba as well. Unfortunately, the time constraints during rebuttal limit this exploration. We will explore our method on other architectures in future work.
>
> [a] COMPOSITIONAL ENTAILMENT LEARNING FOR HYPERBOLIC VISION-LANGUAGE MODELS. ICLR2025.
> [b] Hyperbolic Contrastive Learning for Visual Representations beyond Objects. CVPR2023.
> [c] Rethinking the compositionality of point clouds through regularization in the hyperbolic space. Neurips 2022.
> [d] Hyperbolic Fine-tuning for Large Language Models. ICML2024.

---

### Author Response · Authors · 2025-08-05

Dear Reviewers,

May we kindly ask if our responses have addressed your concerns? We look forward to further discussions and feedback from you!

Sincerely,

Authors

---

### Note · Authors · 2025-08-16

Dear SACs, ACs, and Reviewers,

We sincerely thank you for your invaluable guidance and support throughout the review process. We are deeply grateful to the reviewers for their careful reading and constructive feedback, and to the ACs/SACs for the excellent coordination. The comments and discussions have significantly improved the clarity, rigor, and impact of our work.

**Core Contribution:**

This paper introduces HyperET, a novel paradigm for multi-modal large language models (MLLMs) designed to address the **granularity mismatch** between visual and textual modalities. *(All four reviewers: insightful, interesting and novel)*.

- **High Efficiency**: We propose three parameter-efficient learnable matrix designs (diagonal, block-diagonal, banded) that yield significant improvements with a negligible computational footprint and less than 1% additional parameters. *(All four reviewers).*
- **Robust Generalization**: HyperET's versatility is a key strength. It consistently boosts performance across a wide range of visual encoders (e.g., CLIP, DINOv2, SAM) and foundational MLLM architectures (e.g., Transformer, Mamba). *(Reviewers ijbW, g5HU).*
- **Theoretical Guarantees**: Our approach is grounded in solid theory. We formalize the connection between the hyperbolic radius and granularity, offering a principled and quantifiable solution to an abstract challenge, validated by clear mathematical proofs (Theorems 1 & 2). *(Reviewers ijbW, 1nn1, hmUq).*

**Rebuttal and Revisions:**

The comments and discussions during the review process have helped clarify our contributions and further strengthen the work. We are pleased that our responses successfully **resolved most concerns of all four reviewers**. This has resulted in a consensus of support for our work, with **consistently positive scores.**

We will incorporate all experiments, analyses, and modifications into the final version, and we believe that these revisions, guided by the reviewers’ constructive suggestions, have substantially improved the overall quality of the manuscript and enhanced its potential contribution to the community.

Thank you again for your time and constructive feedback.

Best regards,

Authors of Submission 5784

---

### Decision · Program_Chairs · 2025-09-17

**Decision:**

Accept (oral)

**Comment:**

This paper provides a solution to the granularity mismatch problem between visual and text modalities in MLLMs by leveraging the natural structure of hyperbolic space to align features at various granularities, when training multimodal LLMs. Arbitrary granularity levels can be used by dynamically adjusting the hyperbolic radius using a learnable matrix with Möbius multiplication operations. The authors demonstrate strong performance benefits by conducting comprehensive experiments on both pre-training and fine-tuning scenarios, and particularly reduce object hallucination.

Strengths
- All reviewers find the granularity mismatch problem is well motivated, and the hyperbolic space solution proposed in the paper to be novel, and elegantly grounded in theory, and well supported by clear mathematical proofs.
- All reviewers also find the empirical evaluation setup to be very strong.

Weaknesses
- While many weaknesses have been stated, the authors have addressed all concerns via the rebuttal process and all reviewers have found the answers to be satisfactory. Some of these are 1) Empirical validation that the hyperbolic radius corresponds to different granularities of semantic features, 2) stating efficiency in terms of FLOPs, 3) Demonstrating improvement on MAMBA architectures and not just transformer models, 4) ablations on types of matrix configurations, 5) dynamic choice of granularity per input.

Overall, the paper has received accept scores from all reviewers. The paper is a strong new idea, that is well motivated, works well compared with very strong baselines, is theoretically grounded and is also highly interpretable in its mechanics. Recommend accepting this paper.